# Obesity, Fat Distribution and Risk of Cancer in Women and Men: A Mendelian Randomisation Study

**DOI:** 10.3390/nu14245259

**Published:** 2022-12-09

**Authors:** Nellie Y. Loh, Wenyi Wang, Raymond Noordam, Constantinos Christodoulides

**Affiliations:** 1Oxford Centre for Diabetes, Endocrinology and Metabolism, Radcliffe Department of Medicine, University of Oxford, Oxford OX3 7LE, UK; 2Department of Human Genetics, Leiden University Medical Center, 2300 RC Leiden, The Netherlands; 3Department of Internal Medicine, Section of Gerontology and Geriatrics, Leiden University Medical Center, 2300 RC Leiden, The Netherlands; 4NIHR Oxford Biomedical Research Centre, OUH Foundation Trust, Oxford OX3 7LE, UK

**Keywords:** Mendelian randomisation, obesity, fat distribution, cancer, GWAS

## Abstract

Obesity and upper-body fat distribution are independent, cardiometabolic risk factors but whether they also display comparable associations with cancer risk is unknown. We investigated the causal relationships between body mass index (BMI) and BMI-adjusted waist-to-hip ratio (WHRadjBMI) and cancer risk and searched for potential drivers linking these traits to carcinogenesis using two-sample and multivariable Mendelian randomisation. In women, genetically instrumented higher BMI was associated with lower breast (OR = 0.87, 95% CI 0.81–0.93) and higher endometrial (OR = 1.75, 95% CI 1.55–1.96) cancer risk whilst WHRadjBMI was associated with higher colon cancer risk (OR = 1.22, 95% CI 1.07–1.42). In men, elevated BMI was associated with lower prostate cancer risk (OR = 0.91, 95% CI 0.85–0.98). Mechanistically, testosterone and insulin mediated 21% and 35%, respectively of the total, genetically determined association of BMI with endometrial cancer risk whilst HDL cholesterol and IGF-1 mediated 40% and 22%, respectively of the association between BMI and breast cancer risk. In men, testosterone mediated 21% of the association between BMI and prostate cancer risk. Colon cancer aside, the total amount of body fat might be more important than its location in modulating cancer susceptibility due to differential effects of obesity and fat distribution on adiposity-associated cancer drivers.

## 1. Introduction

According to the World Health Organisation in 2016, 39% of adults over the age of 18 worldwide were overweight and 13% were obese (https://www.who.int/health-topics/obesity, accessed on 1 October 2022) [1]. Obesity increases the risk for many chronic conditions including type 2 diabetes (T2D) and cardiovascular disease (CVD) [2]. Observational studies have also highlighted robust associations between body mass index (BMI) and certain cancers particularly, higher risk of post-menopausal breast, endometrial, and colorectal (CRC) cancers [3,4]. Furthermore, it was estimated that in 2019 around 4.6% of cancer-related deaths in adults and children were attributable to overweight and obesity, which was equivalent to around 462,550 deaths globally [5].

Fat is stored in several depots including subcutaneous abdominal, gluteofemoral, and intra-abdominal (visceral) adipose tissue (AT), which originate from distinct fat progenitors that give rise to specialised adipocytes with diverse functional traits [6,7]. Additionally, visceral obesity is a hallmark of dysfunctional subcutaneous AT, dysregulated adipose cytokine (adipokine) secretion, and insulin resistance [8]. Indeed, whereas upper-body fat distribution is associated with an increased risk of T2D and CVD, gluteofemoral fat accumulation provides protection against cardiometabolic disorders [7]. However, presently, there is limited evidence that surrogate measures of central obesity such as waist-to-hip ratio (WHR) are more strongly and/or differentially associated with cancer risk than BMI [9,10].

The cellular and molecular underpinnings of the connection between adiposity traits and cancer are complex and incompletely understood. Furthermore, given that the associations between measures of adiposity and different cancers reported in observational studies were sex-, site-, and histological subtype-specific, it is unlikely that there is an underlying common set of AT-derived cancer drivers. To date, four main mechanisms have been proposed to link adiposity and cancer [9,11,12]. These include (i) insulin and IGF-1 signalling, (ii) adipokines, the most studied being leptin and adiponectin, (iii) altered sex hormone levels, and (iv) subclinical inflammation. 

Mendelian randomisation (MR) is an epidemiological tool using genetic data to estimate the non-confounded relationships between exposures and outcomes [13]. Herein, we leveraged data from the largest publicly available genome-wide association studies (GWAS) of cancer, adiposity-related anthropometric traits, and proposed cancer drivers to perform two-sample and multivariable MR. Our aim was to comprehensively investigate and contrast the causal relationships between obesity and upper-body fat distribution and cancer risk in women and men and to identify possible mediators linking these traits to carcinogenesis. We chose to study breast, reproductive tract, lung, and colon cancers because this group includes the most common cancer types worldwide (Worldwide cancer data | World Cancer Research Fund International (wcrf.org, accessed on 30 September 2022)) [14], as well as, representing some of the commonest cancers associated with overweight and obesity (https://www.cdc.gov/cancer/obesity/index.htm, accessed on 30 September 2022) [15]. Furthermore, we focused on the effects of BMI and BMI-adjusted WHR (WHRadjBMI) on cancer risk, because WHR associated genetic signals were shown to capture less distinct biology than BMI or WHRadjBMI associated variants [16]. More specifically, whilst GWAS signals associated with WHR were shown to substantially overlap with BMI or WHRadjBMI loci, no overlap between BMI and WHRadjBMI GWAS associated variants was observed. Furthermore, whereas WHR loci were not enriched for gene expression in any tissue or cell type, in contrast, BMI and WHRadjBMI signals were enriched for tissue-specific gene expression predominantly in the central nervous system and adipose tissue, respectively [16].

## 2. Methods

We conducted two-sample MR to investigate associations between BMI and WHRadjBMI, and risk of reproductive tract, lung and colon cancers in men and women of European descent (Appendix A). Where possible analyses using sex-specific exposure instruments and outcome datasets were undertaken. As exposure instruments for adiposity traits, we utilised previously identified, GWAS-significant, independent signals [17] (https://github.com/lindgrengroup/fatdistnGWAS accessed on 1 October 2022). As SNP-outcome data, European population-specific summary-level statistics for cancer traits were obtained from the largest publicly available GWAS (Appendix A). Two-sample MR was also used to examine the relationships between adiposity traits and potential cancer drivers using publicly available, European population-specific GWAS summary statistics (Appendix A) (bioavailable testosterone (BioT) and oestradiol [18], fasting plasma insulin [19], leptin [20], and adiponectin [21]). In the case of IGF-1 [22], HDL cholesterol, triglycerides and high-sensitivity C-reactive protein (hsCRP), where sex-stratified GWAS summary statistics were not publically available, we conducted sex-specific GWAS in European ancestry participants from the UK Biobank (UKBB). The UKBB cohort is a prospective cohort with 502,628 participants between 40–69 years of age recruited from the general population at multiple assessment centres across the UK between 2006 and 2010 [23]. GWAS for IGF-1, HDL cholesterol, triglycerides and hsCRP were conducted in individuals of European descent using linear mixed-model regression analyses implemented in the program BOLT_LMM (version 2.3.2) [24]. IGF-1, HDL cholesterol, triglycerides and hsCRP were rank-based standardised (mean = 0, SD = 1). We adjusted the analyses for age and the first 10 principal components and corrected for the genetic correlation matrix (to correct for familial relationships in the UKBB population). Analyses were done on autosomal chromosomes only. Single nucleotide polymorphisms (SNPs) with minor allele frequency of < 0.01, and those with imputation quality of below 0.3 were excluded. Finally, we interrogated the relationship between adiposity-linked cancer drivers and cancer risk. As genetic instruments, we selected all published independent SNPs for each trait, with a *p*-value < 5 × 10^−8^ (Appendix A). In the case of IGF-1, HDL cholesterol, triglycerides and hsCRP, to obtain a set of independent SNPs for each trait as genetic instruments for MR analyses, we selected all variants with P_BOLT_LMM < 5 × 10^−8^ and performed LD clumping using ld_clump, with the default set at LD r^2^ < 0.001 and a genetic distance of 10 Mb. Due to the small number of GWAS-significant SNPs for fasting insulin (3 for women and 2 for men), we used as genetic instruments the 19 SNPs established for fasting insulin by MAGIC [19]. To minimize risk of weak instrument bias, we selected SNPs with F-statistic > 10.

We employed the inverse-variance weighted (IVW) approach for two-sample MR [25], which assumes that all independent instruments included in the analyses are valid, affect the outcome only through the exposure and do not associate with any confounders. Since this is often not the case, we also performed sensitivity analyses which are more robust to potential pleiotropy of the genetic variants used in the analyses, including the MR-Egger regression [26], weighted-median estimator [27], and MR-PRESSO (MR Pleiotropy RESidual Sum and Outlier) [28]. Bonferroni correction for multiple testing was applied, based on assessments of two anthropometric traits, multiple cancer types (5 for women, 3 for men) and 5 classes of mediators (sex hormones, insulin-IGF-1 axis, lipids, inflammation, adipokines). Given the high correlation between some traits, this level of correction provides a balance between rigorous results and avoiding false negatives. A statistically significant IVW result coupled with directionally consistent associations from all 3 sensitivity analyses was considered as sufficient evidence to claim a causal effect. 

MR based mediation analyses were used to investigate the extent to which cancer drivers mediate the relationship between obesity and cancer in women and men. Mediation analyses were undertaken only if the primary MR analyses involving the adiposity trait, mediator of interest and cancer outcome were significant after Bonferroni correction, with directionally consistent associations in all 3 sensitivity analyses. For this, GWAS summary statistics for BMI (*n* = 171,977 women and 152,893 men of European ancestry) from the GIANT consortium [29] that do not include samples from the UKBB, were used to minimise bias from sample overlap. Exposure data for BMI and mediator of interest were extracted for multivariable MR using mv_extract_exposures_local() in the TwoSampleMR package (v0.5.6). Two-sample MR analyses were performed using GWAS-significant SNPs for BMI and mediator instruments to obtain the total effect of each exposure on the cancer outcome. Multivariable MR was subsequently performed to estimate the direct effect of BMI when adjusted for the effect of the mediator and vise versa. The proportion of the total, genetically determined association of adiposity traits mediated by cancer drivers was calculated by dividing the indirect effect by the total effect. All MR and multivariable MR analyses were conducted using the TwoSampleMR package (v0.5.6) implemented in R (v4.1.0) statistical software [28,30,31].

## 3. Results

### 3.1. MR Analyses in Women

*Associations between adiposity traits and cancer risk.* A 1-SD increase in genetically instrumented BMI was associated with reduced risk of overall (OR = 0.87, 95% CI 0.81–0.93) (Figure 1A, Appendix A), ER positive (ER+) (OR = 0.88, 95% CI 0.82–0.95) and ER negative (ER-) (OR = 0.84, 95% CI 0.75–0.94) breast cancer (Figure 1A, Appendix A). Conversely, genetically determined higher BMI was associated with elevated risk of endometrial cancer (OR = 1.75, 95% CI 1.55–1.96), and a nominally higher risk of lung cancer (OR = 1.19, 95% CI 1.05–1.35) (Figure 1A, Appendix A). Sensitivity analyses revealed directionally uniform associations with IVW MR (Appendix A). Contrasting these findings, genetically instrumented higher WHRadjBMI was only associated with higher colon cancer risk (OR = 1.23, 95% CI 1.08–1.42) (Figure 1A, Appendix A). However, this association was not directionally consistent in the MR-Egger analysis (Appendix A).

*Associations between adiposity traits and potential cancer mediators.* In women, genetically determined raised BMI was associated with higher plasma BioT (β = 0.22 SD, 95% CI 0.19–0.25, *p* = 5 × 10^−53^), fasting insulin (β = 0.19 SD, 95% CI 0.16–0.22, *p* = 1 × 10^−30^), triglycerides (β = 0.27 SD, 95% CI 0.22–0.31, *p* = 1 × 10^−29^), hsCRP (β = 0.48 SD, 95% CI 0.42–0.54, *p* = 1 × 10^−50^) and leptin (β = 0.41 SD, 95% CI 0.36–0.46, *p* = 1 × 10^−62^) levels but lower circulating HDL cholesterol (β = −0.36 SD, 95% CI [−0.42] - [−0.31], *p* = 4 × 10^−38^) and IGF-1 (β = −0.83 SD, 95% CI [−1.08] - [−0.58], *p* = 7 × 10^−11^) concentrations (Figure 1B, Appendix A). Directionally consistent associations to those identified with BMI were also detected between genetically instrumented WHRadjBMI and plasma BioT (β = 0.10 SD, 95% CI 0.07–0.13, *p* = 4 × 10^−10^), fasting insulin (β = 0.16 SD, 95% CI 0.13–0.18, *p* = 5 × 10^−28^), HDL cholesterol (β = −0.38 SD, 95% CI [−0.43] - [−0.34], *p* = 1 × 10^−62^), and triglycerides (β = 0.47;SD, 95% CI 0.42–0.52, *p* = 3 × 10^−76^) (Figure 1B, Appendix A). On the other hand, no association between WHRadjBMI and hsCRP was present whilst associations between WHRadjBMI and circulating IGF-1 (β = 0.42 SD, 95% CI 0.22–0.61, *p* = 4 × 10^−5^) and leptin (β = −0.07 SD, 95% CI [−0.11] - [−0.02], *p* = 0.003) were directionally opposite to those identified with BMI (Figure 1B, Appendix A). Finally, higher WHRadjBMI was associated with lower circulating adiponectin levels (β = −0.25 SD, 95% CI [−0.29] - [−0.21], *p* = 8 × 10^−36^) (Figure 1B, Appendix A). Results were robust to sensitivity analyses (Appendix A).

*Associations between proposed cancer drivers and cancer risk.* A 1-SD increase in genetically instrumented BioT was associated with a 20% higher risk of ER+ breast cancer (OR = 1.20, 95% CI 1.09–1.33) and a 64% elevated risk of endometrial cancer (OR = 1.64, 95% CI 1.41–1.92) (Figure 1C, Appendix A). Consistent results were obtained in MR analyses with genetic instruments modelling the sex hormone binding globulin (SHBG)-independent effects of BioT (Appendix A). Furthermore, genetically determined elevated HDL cholesterol was associated with a higher risk of overall (OR = 1.07, 95% CI 1.03–1.12) and ER- (OR = 1.11, 95% CI 1.04–1.17) breast cancer, as well as a nominally higher risk of ER+ breast cancer (OR = 1.07, 95% CI 1.02–1.12). Suggestive associations were also detected between elevated circulating IGF-1 and higher overall (OR = 1.01, 95% CI 1.00–1.02) and ER+ (OR = 1.01, 95% CI 1.00–1.02) breast cancer risk. Finally, higher fasting insulin was associated with an elevated risk of endometrial cancer (OR = 3.71, 95% CI 2.06–6.67) (Figure 1C, Appendix A) and using an exposure instrument generated from sex-combined data, higher plasma adiponectin was associated with diminished colon cancer risk (OR = 0.50, 95% CI 0.31–0.81). Sensitivity analyses produced uniform associations with IVW MR apart for the association between adiponectin and colon cancer risk (Appendix A).

### 3.2. Mediation Analyses in Women

The negative association between BMI and overall, as well as ER+ breast cancer risk was attenuated after adjustment for HDL cholesterol or IGF-1, which influenced the risk of these cancers independently of obesity. The BMI-adjusted estimates of HDL cholesterol and IGF-1 for overall breast cancer risk were OR 1.07 (95% CI 1.02–1.13) and OR 1.02 (95% CI 1.00–1.04), respectively and for ER+ breast cancer, OR 1.07 (95% CI 1.00–1.14) and OR 1.02 (95% CI 1.00–1.04). Similarly, the association between BMI and endometrial cancer risk was attenuated when adjusted for BioT or fasting insulin (Figure 1D, Appendix A), which influenced the risk of this cancer independent of obesity (BMI-adjusted BioT OR 1.91 [95% CI 1.47–2.48] and insulin OR 2.61 [95% CI 1.44–4.74]). Based on the same analyses, lower IGF-1 and HDL cholesterol were estimated to account for ~22% and ~40%, respectively of the total, genetically determined protective effect of BMI on overall breast cancer risk and ~59% and ~22%, respectively of the protective effect on ER+ positive breast cancer risk. Furthermore, BioT and insulin mediated ~21% and ~35% correspondingly of the total association between BMI and endometrial cancer risk.

### 3.3. MR Analyses in Men

*Associations between adiposity traits and cancer risk.* A 1-SD increase in genetically determined BMI was only associated with a 9% lower prostate cancer risk (OR = 0.91, 95% CI 0.85–0.98), which was consistent in direction and magnitude in sensitivity analyses (Figure 2A, Appendix A). WHRadjBMI was not associated with altered cancer risk in men (Figure 2A, Appendix A).

*Associations between adiposity traits and potential cancer mediators.* In men, genetically instrumented higher BMI was associated with elevated fasting insulin (β = 0.22 SD, 95% CI 0.19–0.26, *p* = 1 × 10^−33^), triglyceride (β = 0.17 SD, 95% CI 0.11–0.24, *p* = 7 × 10^−8^), hsCRP (β = 0.33 SD, 95% CI 0.25–0.41, *p* = 4 × 10^−15^), leptin (β = 0.35 SD, 95% CI 0.31–0.40, *p* = 7 × 10^−50^), and oestradiol (β = 0.01 SD, 95% CI 0.01–0.02, *p* = 0.0003) levels (Figure 2B, Appendix A). Conversely, obesity was negatively associated with plasma BioT (β = −0.13 SD, 95% CI [−0.16] - [−0.10], *p* = 1 × 10^−14^) and HDL cholesterol (β = −0.32 SD, 95% CI [−0.37] - [−0.27], *p* = 7 × 10^−31^). Like BMI, elevated WHRadjBMI in men was associated with higher serum hsCRP (β = 0.10 SD, 95% CI 0.03–0.17, *p* = 0.005) and triglycerides (β = 0.21 SD, 95% CI 0.11–0.30, *p* = 2 × 10^−5^) (Figure 2B, Appendix A). In contrast to obesity however, upper-body fat distribution was not associated with altered BioT, insulin, HDL cholesterol or leptin levels and displayed a diametrically opposite association with serum oestradiol (β = −0.03 SD, 95% CI [−0.05] - [−0.01], *p* = 0.001) (Figure 2B, Appendix A). Finally, elevated WHRadjBMI was negatively associated with plasma adiponectin (β = −0.17 SD, 95% CI [−0.25] - [−0.08], *p* = 1 × 10^−4^). Apart from the association between WHRadjBMI and hsCRP, results were consistent in sensitivity analyses (Appendix A).

*Associations between proposed cancer drivers and cancer risk.* A 1-SD increase in genetically instrumented BioT was associated with a 23% (OR = 1.23, 95% CI 1.14–1.32) higher risk of prostate cancer. Similar results in both direction and magnitude were obtained in MR analyses with genetic instruments modelling the SHBG-independent effects of BioT (Appendix A). Conversely higher circulating leptin was negatively associated with prostate cancer risk (OR = 0.61, 95% CI 0.46–0.81) (Figure 2C, Appendix A). Results were robust to sensitivity analyses (Appendix A).

### 3.4. Mediation Analyses in Men

The negative association between BMI and prostate cancer risk was attenuated after adjustment for BioT or leptin (Figure 2D and Appendix A). Furthermore, BioT influenced the risk of this cancer independent of obesity whilst a similar trend was detected for leptin. Hence, after BMI adjustment, BioT was positively associated (OR = 1.32, 95% CI 1.13–1.55) whilst leptin tended to be negatively associated (OR = 0.75, 95% CI 0.56–1.00) with prostate cancer risk. Finally, we estimated that the reduction in BioT and increase in leptin plasma concentrations mediated ~21% and ~54%, respectively of the total, genetically determined association between BMI and prostate cancer risk.

## 4. Discussion

Herein, we demonstrate that, colon cancer apart, the total amount of AT appears to be more important than its location in influencing cancer susceptibility in both sexes, at least for the cancer types investigated in this study. In contrast, higher WHRadjBMI was equally strongly associated with insulin resistance phenotypes (elevated fasting insulin, low HDL cholesterol and raised triglycerides) compared to BMI in women although more weakly in men. We further show that the distinct associations between obesity and upper-body fat distribution and cancer risk are probably driven by differential effects on the systemic levels of cancer drivers. This was especially true for circulating adiponectin, leptin, and IGF-1 in women, which displayed diametrically opposite associations with BMI compared to WHRadjBMI. Finally, we provide evidence for sex-dependent differences in the associations between both BMI and WHRadjBMI and the levels of cancer drivers, as well as cancer risk. In particular, higher BMI was associated with elevated plasma BioT in women and lower BioT in men whilst WHRadjBMI was selectively associated with increased colon cancer risk in women. The link between increased central adiposity and higher colon cancer risk might reflect the close anatomic relationship between the gastrointestinal tract and visceral AT, which envelopes the colon, as well as surrounding major internal organs such as the liver and pancreas. In this regard, recent evidence suggests that adipocytes within the tumour microenvironment undergo phenotypic changes and contribute to colon cancer development and progression, e.g., through the secretion of inflammatory cytokines, and growth and angiogenic factors. In addition, adipocytes serve as a fuel source for rapidly proliferating tumour cells [32,33].

Our study highlights a potential protective effect of obesity on both ER+ and ER- breast cancer risk. Another MR study conducted in a smaller but overlapping cohort reported consistent associations between BMI and overall, as well as ER- breast cancer risk [34]. Mechanistically, we demonstrate that reduced circulating IGF-1 and HDL cholesterol levels might contribute to the negative association between obesity and overall, as well as ER+ breast cancer development. The IGF-1-ER+ breast cancer association, which was not robust to multiple testing correction in two-sample MR but was supported by mediation analyses, is consistent with both observational and MR data [35]. Furthermore, this link might account for the lack of association between android fat distribution and ER+ breast cancer risk since, in stark contrast to obesity, higher WHRadjBMI was associated with elevated rather than lower circulating IGF-1. The positive association between HDL cholesterol and risk of breast cancer reported herein is consistent in direction and magnitude with the findings of two earlier MR studies exploring the relationship between circulating lipids and breast cancer susceptibility [36,37]. Likewise, an epidemiological study reported that serially measured HDL cholesterol levels were associated with a higher risk of invasive breast cancer [38] whilst in vitro, HDL lipoproteins prepared from the plasma of healthy donors stimulated the proliferation of breast cancer cell lines [39]. Finally, we detected an association between plasma BioT, which was elevated in obesity (Appendix A) and higher ER+ breast cancer risk. This link might reflect direct breast cancer promoting actions of BioT [40] and/or pro-carcinogenic effects of oestradiol since BioT can be converted to oestradiol in extra-gonadal tissues through the action of aromatase.

Our results are consistent with observational data, which have highlighted an inverse association between obesity and breast cancer risk in pre-menopausal women, in keeping with a protective role of increased adiposity on pre-menopausal breast cancer risk [3,10,41,42]. However, they conflict epidemiological findings from several studies and large meta-analyses, which have reported positive associations between obesity and breast cancer risk in postmenopausal women [3,10,42]. This discrepancy might be because genetically instrumented BMI could be a surrogate for early onset obesity [34] and thus not account for post-menopausal weight gain. The latter results in increased systemic exposure to BioT, as well as oestrogen, since oestradiol is predominantly produced in AT via aromatisation of androgens in post-menopausal women. Furthermore, *CYP19A1* expression, encoding aromatase, as well as aromatase activity are amplified in obesity [43,44]. Indeed, although oestradiol levels diminish after the menopause, they have been shown to be higher in obese compared to lean post-menopausal women [45,46]. In contrast, studies in pre-menopausal women have found significantly lower oestradiol levels in subjects who are obese or overweight [46]. This might be due to irregular menstruation and fewer ovulatory cycles in obese and overweight, premenopausal, women due to hyperandrogenism (see above). It has also been postulated that cumulatively higher oestrogen production from the AT as well as the ovaries of obese pre-menopausal women would result in diminished gonadotrophin secretion, due to suppression of the hypothalamo-pituitary axis via negative feedback, in turn leading to reduced ovarian function and amenorrhoea [47]. An additional explanation could be that compared to early onset obesity, which is associated with hyperplastic AT expansion characterised by an increased adipocyte number, weight gain during adulthood is associated with hypertrophic obesity characterised by enlarged adipocytes [48,49]. Adipocyte hypertrophy is associated with AT dysfunction and insulin resistance [50], which, e.g., suppresses the serum concentration of SHBG thereby increasing the systemic concentrations of bioavailable oestrogen and testosterone. Aromatase expression in human breast adipose stromal cells was also shown to be potently induced by pro-inflammatory cytokines, which are secreted by hypertrophic adipocytes [42]. Finally, another potential contributor to the differences between genetic and observational data is confounding of observational studies by unmeasured variables, for example lifetime exposure to environmental carcinogens such as alcohol [39] or weight promoting effects of chemotherapy in breast cancer patients [51].

Another finding of this study was a strong association between obesity and endometrial cancer susceptibility, which is in line with observational data [3,10,52]. For example, for every 5-units increase in BMI (~1 BMI SD in this study) the risk of uterine cancer was shown to increase by 54% [10]. This might reflect endometrial exposure to oestrogen unopposed by progesterone consequent to anovulatory menstrual cycles in premenopausal obese women [45,46,53] and/or exposure to higher and unopposed systemic oestradiol levels in obese post-menopausal women (see above). Based on prospective epidemiological data, the risk of endometrial cancer was >2-fold higher in women with the highest circulating oestrogen levels [54]. Furthermore, oestrogen unopposed by progesterone, e.g., during oestrogen-only hormone replacement therapy was associated with a 2.3-fold increased endometrial cancer risk [55,56]. In accordance with these findings, we demonstrate both a positive association between BioT and endometrial cancer susceptibility, as well as identifying BioT as a mediator of the effect of genetically instrumented BMI on increased endometrial cancer risk. BioT might also have direct effects on endometrial cancer development as also suggested by epidemiological data [54]. Fasting insulin was also strongly and positively associated with elevated endometrial cancer susceptibility in two-sample MR with mediation analyses further highlighting hyper-insulinaemia as a potential driver of the obesity-endometrial cancer link. This latter result is broadly consistent with data from the cancer genome project showing that ~90% of endometrial cancers harbour insulin signalling pathway mutations (www.cbioportal.org accessed on 1 October 2022). A previous meta-analysis of observational studies similarly reported a higher endometrial cancer risk in women with raised fasting insulin [57]. 

Contrary to obesity, upper-body fat distribution was associated with a 22% higher colon cancer risk in women. However, this result was based on a small number of cases (*n* = 2226) and was inconsistent in sensitivity analyses. As such, it should be interpreted with caution. Another MR study in a large cohort [58] investigated the associations between adiposity traits and CRC risk and showed that elevated BMI was associated with a higher risk of CRC in men than women. In contrast, higher WHR was more strongly associated with CRC risk among women (IVW OR 1.25 versus 1.05). Likewise, epidemiological studies have revealed that obesity is more strongly associated with colon cancer risk in men than women [10]. Consistent, but less robust, sexually dimorphic associations were also reported between WHR and colon cancer risk [10] contrasting the MR findings highlighted above [58]. Mechanistically, hypo-adiponectinaemia and possibly raised IGF-1 levels might drive the WHRadjBMI-colon cancer risk connection. Notably, both adiponectin and IGF-1 displayed directionally opposite associations with upper-body fat distribution compared to obesity in females. However, the relationship between adiponectin and colon cancer was not directionally consistent in sensitivity analyses whilst the association between IGF-1 and colon cancer did not survive multiple testing correction. Whilst tentative, our finding of a potential causal genetic relationship between hypo-adiponectinaemia and elevated colon cancer risk is in keeping with data from both observational and animal studies [12]. Prospective epidemiological and MR study data likewise highlighted a positive association between circulating IGF-1 and CRC risk [59]. 

Lastly, we identify a negative association between obesity and prostate cancer risk, which is consistent in direction and magnitude with the 6% lower risk of localised prostate cancer per 5-units increase in BMI reported by an umbrella meta-analysis of epidemiological studies [10]. Mechanistically, we demonstrate that higher BioT is associated with an elevated risk of prostate cancer and based on mediation analyses that reduced BioT levels partially drive the diminished prostate cancer risk in obese men. Our findings contrast the lack of firm epidemiological data linking serum concentrations of BioT and the risk of developing prostate cancer. In this regard, a recent pooled analysis of 20 prospective studies revealed that only men in the lowest decile of free testosterone concentration had a diminished risk of overall prostate cancer compared to men with higher concentrations [60]. Another potential driver of the negative association between obesity and prostate carcinogenesis might be hyper-leptinaemia, which based on mediation analyses might account for up to half of the total, genetically determined, protective effect of BMI on prostate cancer risk. This result was unexpected because whilst leptin is thought to play a pivotal role in linking obesity to cancer development, it is generally believed to have pro-tumourigenic effects [11,12]. However, a recent meta-analysis of prospective epidemiological studies found a weak, negative association between plasma leptin and prostate cancer risk [61]. Nonetheless, the same report also highlighted a directionally opposite association between leptin and risk of prostate cancer based on meta-analysis of cross-sectional data, as well as describing a weak, positive link between circulating leptin levels and aggressive disease.

Our findings should be interpreted in the context of some limitations. Firstly, we used GWAS summary statistics for anthropometric, rather than imaging, measures of adiposity traits. Secondly, our results/conclusions are based on data from GWAS conducted in European ancestry subjects and thus might not extend to other ethnic populations. Thirdly, the summary data that we leveraged for colon cancer were generated from a small UKBB subsample increasing the likelihood of false-negative associations. Fourthly, the genetic instruments for leptin, adiponectin, and fasting insulin, explained only a small fraction of the genetic variance in these traits, which, might have resulted in low statistical power and missed associations between these traits and certain cancers. Nonetheless, because all the exposure instruments pertaining to these phenotypes had an F-statistic > 10, the risk of weak instrument bias is negligible. Another study limitation is that the genetic instruments for lung and colon cancer, as well as adiponectin and leptin were derived from sex-combined data. IGF-1 circulates mostly bound to IGF-binding proteins with only free IGF-1 being bioactive. However, the results reported here, reflect total IGF-1 levels and thus likely underestimate any true pro-tumourigenic effects of this growth factor. Finally, we modelled genetic but not environmental associations.

In summary, whilst both obesity and upper-body fat distribution are major and independent metabolic and cardiovascular risk factors, the total amount of body fat seems to be more important than its location in modulating cancer susceptibility. Additionally, obesity and upper-body fat distribution display different and occasionally directionally opposite associations with adiposity-linked cancer drivers. Our data also shed light on the role of proposed cancer drivers in mediating the cancer modulating effects of adiposity. In particular, we highlight BioT, as well as HDL cholesterol and IGF-1 as potential drivers linking excess AT with altered endometrial and breast cancer susceptibility, respectively in women and BioT and potentially leptin as mediators of the protective effect of obesity on prostate cancer risk in men. As such, these should be the focus of future pre-clinical and clinical studies targeted in the treatment of these cancers.

## Figures and Tables

**Figure 1 nutrients-14-05259-f001:**
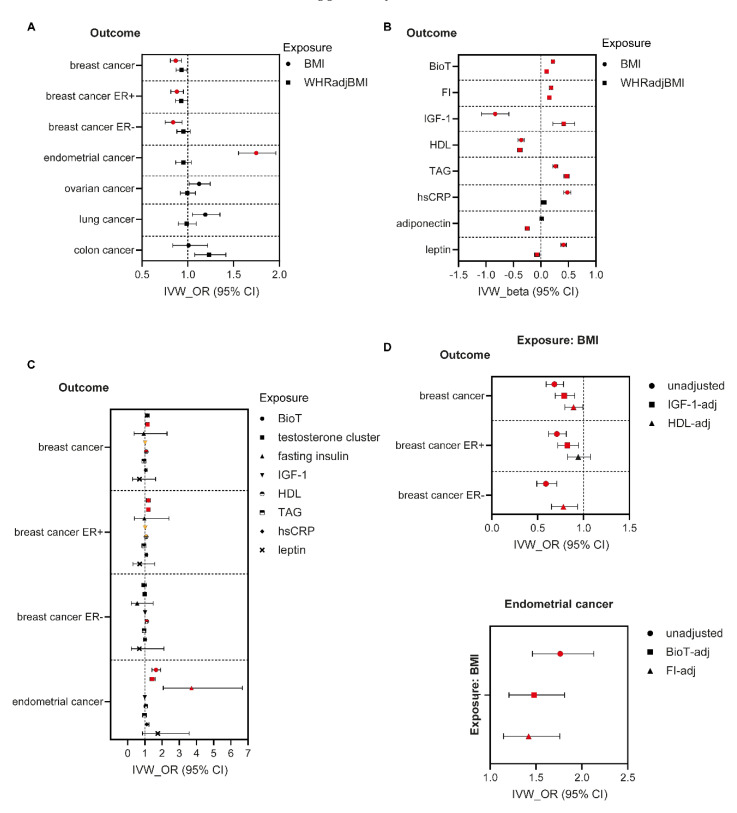
Two sample MR (IVW) estimates of effects of adiposity measures and traditional cancer drivers on cancer risk in women. (**A**) IVW estimates (OR, 95%CI) of effects of BMI and WHRadjBMI on cancer risk in women. (**B**) IVW estimates (beta, 95%CI) of effects of BMI and WHRadjBMI on potential cancer mediators in women. (**C**) IVW estimates (OR, 95%CI) of effects of mediators on cancer risk in women. (**D**) Mediation analyses: IVW estimates of effects of BMI on breast and endometrial cancers, adjusted for indicated mediator.,Red-filled symbols: Analyses yielding IVW results with *p* ≤ 0.005 (**A**), *p* ≤ 0.005 (**B**), and *p* ≤ 0.003 (**C**) (Bonferroni correction for multiple testing) and that are directionally consistent across all sensitivity analyses, and (**D**) *p* < 0.05 with adjustment for mediator. Orange-filled symbols in (**C**): IVW results with *p* ≤ 0.009 and that are directionally consistent across all sensitivity analyses. BioT, bioavailable testosterone; FI, fasting insulin; IGF-1, insulin-like growth factor-1; HDL, high-density lipoprotein cholesterol; TAG, triglycerides; hsCRP, high-sensitivity C-reactive protein; ER, oestrogen receptor.

**Figure 2 nutrients-14-05259-f002:**
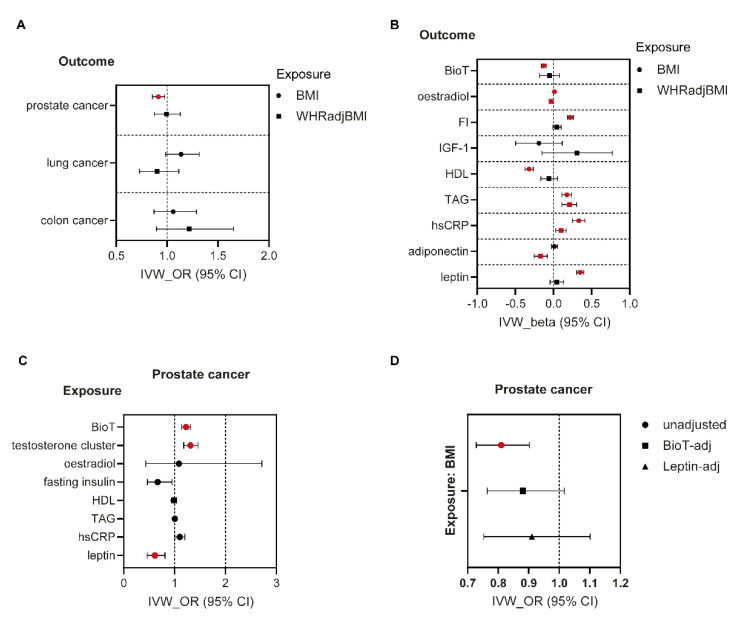
Two sample MR (IVW) estimates of effects of adiposity measures and traditional cancer drivers on cancer risk in men. (**A**) IVW estimates (OR, 95%CI) of the effects of BMI and WHRadjBMI on cancer risk in men. (**B**) IVW estimates (beta, 95%CI) of the effects of BMI and WHRadjBMI on potential cancer mediators in men. (**C**) IVW estimates (OR, 95%CI) of effects of mediators on risk of prostate cancer. (**D**) Mediation analysis: IVW estimates of effects of BMI on prostate cancer, adjusted for indicated cancer mediator. BMI-effect was not significant after adjustment for BioT or leptin. Red-filled symbols: Analyses yielding IVW results with *p* ≤ 0.008 (**A**), *p* ≤ 0.005 (**B**), and *p* ≤ 0.005 (**C**) (Bonferroni correction for multiple testing) and that are directionally consistent across all sensitivity analyses. BioT, bioavailable testosterone; FI, fasting insulin; IGF-1, insulin-like growth factor-1; HDL, high-density lipoprotein cholesterol; TAG, triglycerides; hsCRP, high-sensitivity C-reactive protein.

## Data Availability

Full GWAS summary statistics for the exposure and outcome data used herein can be found at https://www.ebi.ac.uk/gwas [BioT, oestradiol, adiponectin, leptin and leptin (adjBMI)], https://github.com/lindgrengroup/fatdistnGWAS (BMI and WHRadjBMI), https://portals.broadinstitute.org/collaboration/giant/index.php/GIANT_consortium_data_files (BMI), https://magicinvestigators.org/ downloads/ (fasting insulin) and https://gwas.mrcieu.ac.uk (cancer traits). Sex-specific GWAS summary statistics generated from the UKBB (HDL, TAG, hsCRP, IGF-1) are available from the corresponding author upon reasonable request.

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
