# Peer review of "Obesity, Fat Distribution and Risk of Cancer in Women and Men: A Mendelian Randomisation Study"

_nutrients, 2022, doi:10.3390/nu14245259_

Round 1
Reviewer 1 Report
This manuscript is not registered.
This manuscript is of novelty, I searched using "Obesity[TI] AND Fat[TI] AND Cancer[TI]" in PubMed and fail to search the same study. The methodology is overall adequate.
Author Response
This manuscript is of novelty, I searched using "Obesity[TI] AND Fat[TI] AND Cancer[TI]" in PubMed and fail to search the same study. The methodology is overall adequate.
Thank you.
Reviewer 2 Report
Loh et al., through a mendelian randomisation study, investigated the relationship between obesity and upper-body fat distribution and cancer risk and drivers.
· For website citation (WHO or others..) please use the following mode suggested by the journal: Title of Site. Available online: URL (accessed on Day Month Year).
· Line 36-38: “ it was estimated that in 2012 .. cancer cases globally”, I suggested to look for an updated source if applicable.
· Line 65-66: “ because WHR associated genetic signals were shown to capture less distinct biology than BMI or WHRadjBMI associated variants” what does it mean? please explain better.
· Is there a specific reason for choosing risk of reproductive tract, lung and colon cancers in men and women among other cancer? Why for prostate cancer (PC) the authors excluded the sub-division of PC (androgen deprivation therapy ADT-sensitive or resistance cancer) since they correlated the risk with the testosterone.
· Line 260-262: Visceral fats surround and infiltrate into different organs in the gastrointestinal tracts including also liver and pancreas not only the colon. Please adjust and discuss better
· The “obesity paradox” for prevention or promotion of cancer should be discussed and clarified, especially when talking about the relation of obesity with breast cancer risk that differs in premenopausal and postmenopausal women. Evidence suggests that before the menopause, the level of adiposity is inversely related to risk, indicative of a protective effect, whereas in postmenopausal women, particularly the elderly, the association is a positive one, consistent with obesity being a risk factor
· The manuscript should also include graphical abstract as recommended by the journal guidelines, a table of abbreviations and definitions is also appreciated
· As journal stated that there is no restriction on the length of the papers. The full experimental details must be provided so that the results can be reproduced, authors can add summery table for GWASS database for more clarification for the readers
Round 2
Reviewer 1 Report
publish